# Ongoing Use of SSRIs Does Not Alter Outcome in Hospitalized COVID-19 Patients: A Retrospective Analysis

**DOI:** 10.3390/jcm11010070

**Published:** 2021-12-24

**Authors:** Steven H. Rauchman, Sherri G. Mendelson, Courtney Rauchman, Lora J. Kasselman, Aaron Pinkhasov, Allison B. Reiss

**Affiliations:** 1The Fresno Institute of Neuroscience, Fresno, CA 93730, USA; cbr.courtney@gmail.com; 2Providence Southern California Region, Irvine, CA 92612, USA; Sherri.Mendelson@providence.org; 3Department of Medicine and Biomedical Research Institute, NYU Long Island School of Medicine, Mineola, NY 11501, USA; Lora.Kasselman@NYULangone.org (L.J.K.); Aron.Pinkhasov@NYULangone.org (A.P.); Allison.Reiss@NYULangone.org (A.B.R.)

**Keywords:** COVID-19, disease severity, clinical presentation, SSRI, antidepressant

## Abstract

SARS-CoV-2 continues to have devastating consequences worldwide. Though vaccinations have helped reduce spread, new strains still pose a threat. Therefore, it is imperative to identify treatments that prevent severe COVID-19 infection. Recently, acute use of SSRI antidepressants in COVID+ patients was shown to reduce symptom severity. The aim of this retrospective observational study was to determine whether COVID+ patients already on SSRIs upon hospital admission had reduced mortality compared to COVID+ patients not on chronic SSRI treatment. Electronic medical records of 9044 patients with laboratory-confirmed COVID-19 from six hospitals were queried for demographic and clinical information. Using R, a logistic regression model was run with mortality as the outcome and SSRI status as the exposure. In this sample, no patients admitted on SSRIs had them discontinued. There was no significant difference in the odds of dying between COVID+ patients on chronic SSRIs vs. those not taking SSRIs, after controlling for age category, gender, and race. This study shows the utility of large clinical databases in determining what commonly prescribed drugs might be useful in treating COVID-19. During pandemics due to novel infectious agents, it is critical to evaluate safety and efficacy of drugs that might be repurposed for treatment.

## 1. Introduction

The COVID-19 pandemic has resulted in an unprecedented worldwide response in the form of a multitude of clinical trials designed to develop efficacious prophylactic and therapeutic interventions [1]. The life-threatening global health crisis has encouraged innovation with a focus on rapid results and this has led to the evaluation of existing pharmaceuticals for potential repurposing as COVID-19 treatments [2,3]. The low cost and widespread availability of some drugs already on the market has made them attractive from a social and medical perspective. A number of anti-viral agents are under investigation, but are still pending formal approval in the United States [4,5]. Unfortunately, the true efficacy of some of these compounds has not been supported by more rigorous clinical trials [6].

The common usage of psychotropic drugs in the general population has led to an interest in their effects on the course and mortality of patients diagnosed with COVID-19. Crisanto Diez Quevedo et al. provide one of the most comprehensive studies of psychiatric diagnosis and use of psychotropic drugs before and during hospitalization for COVID-19 [7]. This observational study analyzed 2150 patients hospitalized with COVID-19 with a variety of psychiatric comorbidities who received psychotropic drug treatments of different drug classes, and was not a dedicated SSRI study. In this group, 14% were already on psychopharmacological treatments, while 41.4% were prescribed psychotropic drugs de novo upon or after admission to the hospital. SSRIs were prescribed, but other anti-anxiety drugs were more commonly administered. Mortality rate was 17% and 14% required admission to an ICU. History of depressive disorders, and development of delirium during admission, were associated with higher mortality. They found an association between use of anxiolytics/hypnotics or antidepressants during the previous and reduced mortality, but could not confirm any cause-and-effect relationship.

The potential value of selective serotonin reuptake inhibitors (SSRIs) and selective norepinephrine reuptake inhibitors (SNRIs), typically prescribed for anxiety, depression and obsessive-compulsive disorder (OCD), has been discussed significantly in the scientific literature [8] and lay press [9]. Numerous in vitro studies have carefully delineated multiple inflammatory pathways in which SSRIs and SNRIs might be beneficial in reducing inflammation [10,11,12,13,14]. The key role of inflammation in the progression, morbidity, and mortality of COVID-19 has been well documented in the medical literature and cytokine storm syndrome is a life-threatening feature of severe COVID-19 [8,15,16,17]. Depressed persons often exhibit raised blood levels of proinflammatory cytokines and SSRIs may reduce these levels [18,19]. Anti-inflammatory effects of SSRIs may thus underlie their possible protective role in COVID-19 [11]. Serotonin may have a direct effect on the immune system and COVID-19 may induce serotonin deficiency [20,21]. An example of serotonin’s effect on the immune system is seen in T lymphocytes where SSRIs decrease their viability and immunoreactive protein content [22]. SSRIs have been found to dampen key inflammatory pathways such as the signal transducer and activator of transcription 3 (STAT3) and nuclear factor (NF)–κB pathways [9]. This, in turn, attenuates downstream proinflammatory cytokine expression, notably that of interleukin-6 and tumor necrosis factor (TNF)-α, both of which are implicated in cytokine storm [23,24].

Fluoxetine and other SSRIs might have a direct anti-viral effect [25,26]. Interestingly, it has been found that the COVID-19 virus activates the ceramide system that then facilitates viral entry into cells [27]. The sphingomyelin/ceramide system can be altered by SSRIs so that ceramide levels are reduced and this may prevent COVID-19 replication [28,29,30].

Early in the pandemic, a large French multi-center retrospective study [8] suggested the beneficial role of SSRIs in preventing intubation and death in hospitalized COVID-19 patients. The SSRIs needed to be continued within the first 48 h of hospital admission. The prior use of these drugs in individuals as outpatients before contracting COVID-19 is not clearly described. There was also a noteworthy exclusion of many patients because of incomplete medical records. A key limitation of this ambitious and important review is the sudden inundation of the French health care system with large numbers of very sick COVID-19 patients. The retrospective nature of the study was intended to encourage more rigorous prospective investigations.

There have since been a number of publications on SSRIs and COVID-19, and these have garnered attention from the lay press. Stories have appeared in the Los Angeles Times and CBS News and the subject was featured on the national television news magazine program “60 Minutes”. The importance of exploring the role of SSRIs in COVID was noted in Nature by a co-author of this paper, Steven Rauchman [31]. With the presence of effective vaccines, conducting a large prospective clinical trial of therapeutics in the US, Europe, and other nations with large vaccination programs loses feasibility. Simply stated, those fortunate populations cease being a control or treatment group available for potential therapeutic drugs, yet the need for effective therapeutics to prevent suffering and death among a significant part of the world population remains. Vaccines will not reach many less advantaged nations in time and, with new strains, breakthrough cases are emerging in vaccinated populations [32]. The omicron variant is the most recent to cause concern due to its high transmissibility and large number of mutations in the spike protein, responsible for virus invasion of host cells [33]. 

The purpose of this study is to explore the utility of SSRIs in the setting of acute COVID-19 illness, not only as a means to resolve the issue of its effectiveness, but to provide a paradigm for evaluating repurposed drugs and to address the issue of maintenance medication continuation/discontinuation decisions in the acute care setting.

## 2. Materials and Methods

This study was approved by the Providence Health and Services IRB as a minimal risk study on 31 March 2021. Providence Health and Services IRB is an electronic IRB serving the 52 hospitals and 1085 clinics within the large Providence System located in 7 states along the West Coast. Providence Health and Services IRB is compliant with U.S. Health and Human Services regulations and requires CITI training and conflict of interest attestations for all investigators. All research studies are required to obtain IRB approval or exemption prior to initiation. There are more than 1700 published studies with Providence Health and Services IRB approval. A retrospective observational study design was used. Therefore, it was determined by the IRB that consent would be waived. Electronic medical records of 9044 patients with a laboratory-confirmed diagnosis of COVID-19 from March 2020 to March 2021 from six hospitals were queried for discharge date and disposition; medications on admission including SSRIs/SNRIs; age; gender; ethnicity; admission to ICU; discontinuation of antidepressant medications upon ICU admission, and hospital facility.

Inclusion criteria: adult patients 18 and over admitted with a diagnosis of COVID-19 and on an antidepressant drug during admission. 

Exclusion criteria: patients under 18 years of age, without an admission diagnosis of COVID-19 and not on an antidepressant drug on admission.

Using R version 3.6.2 (R Core Team, 2021, open source, R Foundation for Statistical Computing, Vienna, Austria), a logistic regression model was run with mortality as the outcome and SSRI status as the exposure. Initially, anti-depressant drugs were separated out (SSRI status; SSRIs vs. SNRIs) in the model but since there were no significant differences among the drug classes, they were combined in the final models. An adjusted logistic regression model was run to account for age category, gender, race, and hospital facility. All tests were considered significant at p of 0.05 or less. For inclusion in the statistical model, age category (<18, 18–30, 31–40, 41–50, 51–60, 61–70, 71–80, 81+ years) was recoded into the median year for each age range. One person was dropped from the analysis because their self-reported gender (female, male) was listed as “unknown”. Self-reported race categories were White or Caucasian, Hispanic or Latino, Black or African-American, Asian, Native Hawaiian or Other Pacific Islander, American Indian or Alaska Native, Other, Unknown. Any person who indicated “refused to answer” or had missing race information was categorized as “Unknown” for the analysis. Hospital facility names were Providence Holy Cross Medical Center, Providence Little Company of Mary (San Pedro), Providence Little Company of Mary (Torrance), Providence Saint John’s Health Center, Providence St. Joseph Medical Center (Burbank), and Providence Tarzana Medical Center. 

## 3. Results

Figure 1 is a flow chart of the retrospective cohort study. 

Demographic information on our population is shown in Table 1. Of the 832 patients that continued taking SSRIs/SNRIs, the following were represented: citalopram hydrobromide (*n* = 109), desvenlafaxine (*n* = 1), duloxetine HCl (*n* = 145), escitalopram oxalate (*n* = 227), fluoxetine HCl (*n* = 87), paroxetine HCl (*n* = 48), sertraline HCl (*n* = 175), and venlafaxine HCl (*n* = 40). The odds of dying do not differ significantly in hospitalized COVID+ patients based on whether or not they are taking SSRIs/SNRIs (Table 2). 

There is no significant difference in the odds of dying between COVID+ patients on SSRIs vs. COVID+ patients not taking SSRIs. The odds of COVID+ patients on SSRIs dying is 1.09 (95% CI: 0.91, 1.31) compared to COVID+ patients not on SSRIs (*p* = 0.35). There is no significant difference in the odds of dying between COVID+ patients on SSRIs vs. COVID+ patients not taking SSRIs, after controlling for age category, gender, primary race, and facility location, the odds of COVID+ patients on SSRIs dying are 0.96 (95% CI: 0.79, 1.16) compared to COVID+ patients not on SSRIs (*p* = 0.69; Table 2).

## 4. Discussion

As noted previously, the initial study from France that implicated SSRIs as potential therapeutic tools in COVID-19 was beset by a number of limitations [8]. A few other small studies have supported the results from France. In a randomized prospective clinical trial of 152 outpatients with confirmed COVID-19 given either fluvoxamine or placebo, early clinical introduction of fluvoxamine decreased likelihood of clinical deterioration over a 15-day period [34]. The authors readily acknowledged the difficulty in recruiting patients. The short duration and small sample size were other limiting variables. The authors recommended larger prospective clinical trials.

Another important SSRI investigation took place early in the pandemic at a San Francisco Bay Area racetrack [35]. A large number of racetrack employees were diagnosed with COVID-19 in a very brief period of time. The racetrack physician offered fluvoxamine to the infected racetrack employees and approximately 50% of the employees took the drug while the remainder declined. There was no treatment available for early intervention in COVID-19 at that point in time besides supportive care. According to this study, fluvoxamine prevented serious clinical deterioration and hospitalization. 

A larger prospective randomized clinical trial on fluvoxamine was recently completed in Brazil led by investigators from Brazil and Canada [36]. This study appears to support the value of fluvoxamine in early intervention in COVID-19 patients in preventing progression, serious complications, and mortality. 

A recently published large retrospective study from the University of California, San Francisco, and Stanford University (UCSF/Stanford) suggests that SSRIs may be associated with decreased mortality from COVID [37]. This evaluation of in excess of 80,000 patients found that reduced mortality was confined to those taking fluvoxamine and fluoxetine. Importantly, the authors only considered patients taking SSRIs within a defined time period (10 days before and four days after a diagnosis of COVID-19. Patients with documented SSRI use outside of this window were excluded from the study. This resulted in exclusion of 7250 of the 83,584 patients. In contrast, our retrospective analysis examined a somewhat different patient population, which included all patients taking SSRIs upon hospital admission for presumptive pre-existing psychiatric diagnosis. The underlying assumption under these criteria is that the vast majority had been on SSRIs for a significant period of time. In our study, approximately 10% of COVID-19 admissions were already on SSRIs. This is similar to the rate of use of this class of drugs in the general population. The UCSF/Stanford study included a population in which 4.1% of patients were on SSRIs. Presumably, the UCSF/Stanford study includes patients who were being prescribed SSRIs for a very recent diagnosis of a psychiatric disorder or individuals where SSRIs were added after hospital admission and COVID-19 diagnosis. Our study was not focused on the need for acute intervention with SSRIs to treat patients admitted with COVID-19. 

The initial inquiry by our group was directed at recreating the original French study. The retrospective review presented here is not a double-blind placebo-controlled randomized clinical trial because that would be costly and difficult to execute in the midst of a pandemic. 

A significant segment of the adult population in the United States (estimated at 10–20%) is already taking SSRIs and the rate of antidepressant use has been increasing in the last decade [38]. This supports the need to use available data to determine definitively whether patients who are already taking antidepressants fare better or worse than patients not taking such medications. This is especially crucial in more severe COVID-19 cases requiring hospitalization. Our study indicates that, in a population hospitalized for COVID-19, there was no clinical benefit of SSRIs that were being taken before and during admission. Since use of SSRIs for anxiety, OCD or depression seems unrelated to other comorbidities known to affect COVID morbidity and mortality (hypertension, diabetes, heart disease), direct impact of SSRIs on the risk factors for severe COVID-19 is minimal. There is one important caveat—the weight gain that may accompany use of some SSRIs [39]. 

A limitation of our study is that our patient population did not have any patients taking fluvoxamine and this is in direct contrast to the French study and to an open label, prospective cohort study of fluvoxamine in ICU patients with COVID-19 from Croatia [8,40]. In the Croatian group, 51 COVID-19 patients with severe disease admitted to the ICU were started on fluvoxamine 100 mg three times per day for 15 days in addition to standard therapy. These patients were prospectively matched to 51 controls with COVID-19 who were not given fluvoxamine. Overall mortality was lower in the fluvoxamine group, 58.8% than in the control group, 76.5% (*p* = 0.027).

Our study examined severely ill patients while a recent retrospective observational study from Rockland County, New York, USA, looked at a cohort of 165 long-stay psychiatric facility in-patients and found that those receiving antidepressants had 72% lower odds of testing positive for COVID-19. SSRIs and SNRIs were the drivers of this significantly decreased likelihood of COVID-19 infection. Outcome once infected with COVID-19 was not assessed in this analysis [41]. 

Fluvoxamine has been shown to be a sigma-1 receptor (S1R) agonist with the strongest binding affinity to S1R of all the SSRIs [41]. S1R is a chaperone protein located at the endoplasmic reticulum-mitochondrion interface that regulates autophagy, an important process in viral evasion [42]. It has cytoprotective and anti-inflammatory properties [43]. By activating S1R, fluvoxamine exerts immunomodulatory effects and can reduce cytokine production [44]. Other SSRIs do not activate S1R as potently and the absence of a fluvoxamine subset of patients may have affected our results [45,46]. 

Another important clinical issue is whether SSRIs are continued or discontinued when a COVID-19 patient is hospitalized or admitted to an ICU. SSRIs are frequently discontinued in the ICU patient [42,43]. There is literature showing that discontinuation results in adverse ICU outcomes [42,43]. SSRIs are often inadvertently discontinued because they are not considered important in acute disease. This can cause increased agitation among these patients and need for additional sedation, which may in turn depress respiration. There are sometimes relative contraindications to SSRI use in ICU patients. ECG changes and coagulation issues have been noted [44,45]. The ICU physicians in this study did not routinely discontinue the use of SSRIs in patients unless medically indicated. Hospital policy is to continue antidepressants while patients are hospitalized. The health organization has a Clinical Institutes model that shares expertise and research across the region’s 13 hospitals and affiliated hospitals and this policy was already in existence prior to this study and was not in any manner attributable to this study. As we debate the potential value of such drugs in very sick patients with COVID-19, a collateral effect might be a re-examination of ICU drug protocols for all patients. Among the drug classes with anti-inflammatory properties routinely prescribed in the middle aged and senior populations that merit review are the statins, metformin, and antihypertensives, particularly angiotensin-converting enzyme inhibitors [46,47,48]. Perhaps there is synergistic activity in patients on several of these medications. Large retrospective studies of these commonly used drugs would begin to resolve these issues.

Clearly, there are many aspects (of timing and situation), with regard to SSRI/SNRI use and the impact on COVID-19 outcome that remain unresolved.

## 5. Conclusions

The impact of psychiatric disorders and SSRIs on COVID-19 evolution and mortality has been referenced in the literature with no clear-cut results [49,50]. In the present retrospective study of 9044 patients hospitalized for COVID-19 in six California hospitals of a large hospital system in the Western US, prior use of SSRIs or SNRIs did not reduce mortality. These drugs were continued during hospitalization and had been started prior to the onset of COVID-19, presumably for a pre-existing psychiatric condition. 

This study shows the utility of large clinical databases in addressing the urgent issue of determining what commonly prescribed drugs might be useful in treating COVID-19. The ongoing nature of the pandemic despite the vaccine rollout signals a pressing need to mitigate COVID-19 sequelae and the repurposing of readily available and inexpensive medications has the potential to save lives, particularly because rapid implementation could occur. As a result of this study, the use of the SSRI/SNRI drug class does not hold a particular advantage in the patient population already taking these drugs. Our study does not provide evidence that there is any benefit to keeping patients on SSRIs in the ICU. COVID-19 data are already embedded in medical records and available within electronic health systems; thus, filling in the gaps in knowledge, about the effects on morbidity and mortality of specific SSRIs and drug combinations, should be feasible [51]. 

## Figures and Tables

**Figure 1 jcm-11-00070-f001:**
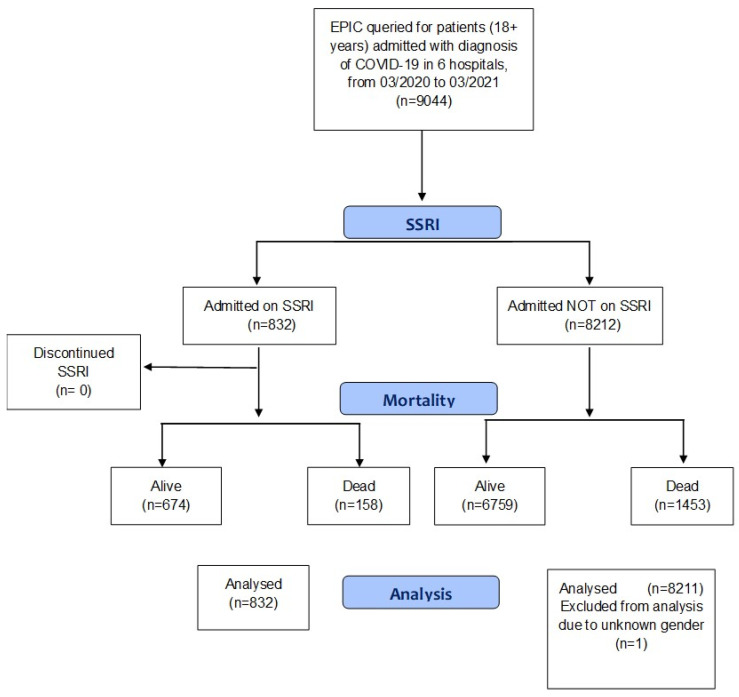
Flow chart of retrospective cohort study of COVID-19 hospitalized patients categorized by use/non-use of SSRIs on admission and subsequent continuation/discontinuation of SSRIs. Scheme depicts study design, cohort selection, and outcome.

**Table 1 jcm-11-00070-t001:** Demographic information.

Demographic		Patients on SSRIs/SNRIs (%)	Patients Not on SSRIs/SNRIs (%)
		*n* = 832	*n* = 8211
Gender			
	Female	504 (60.6)	3646 (44.4)
	Male	328 (39.4)	4565 (55.6)
Age Group			
	>81	248 (29.8)	1623 (19.8)
	71–80	203 (24.4)	1489 (18.1)
	61–70	181 (21.8)	1683 (20.5)
	51–60	93 (11.2)	1458 (17.8)
	41–50	59 (7.1)	799 (9.7)
	31–40	35 (4.2)	663 (8.1)
	18–30	13 (1.6)	455 (5.5)
	<18	0 (0.0)	41 (0.5)
Primary Race			
White or Caucasian		409 (49.2)	2151 (26.2)
Hispanic or Latino		277 (33.3)	4196 (51.1)
Black or African American		43 (5.2)	493 (6.0)
Asian		28 (3.4)	476 (5.8)
Native Hawaiian or Other Pacific Islander		2 (0.2)	60 (0.7)
American Indian or Alaskan Native		0 (0.0.)	5 (0.1)
Other		61 (7.3)	683 (8.3)
Unknown		12 (1.4)	147 (1.8)
Facility Name			
Providence Holy Cross Med Center		2277 (27.7)	186 (22.4)
Providence Little Company of Mary (San Pedro)		450 (5.5)	41 (4.9)
Providence Little Company of Mary (Torrance)		2262 (27.5)	137 (16.5)
Providence Saint John’s Health Center		607 (7.4)	103 (12.4)
Providence St. Joseph’s Medical Center (Burbank)		1622 (19.8)	209 (25.1)
Providence Tarzana Medical Center		993 (12.1)	156 (18.8)

**Table 2 jcm-11-00070-t002:** Odds of death in COVID+ patients on continuation of SSRIs/SNRIs during hospitalization.

Variables	Crude ORs (95% CI)	*p*-Value	Adjusted ORs (95% CI)	*p*-Value
SSRI on admission:				
No	1		1	
Yes	1.09 (0.91, 1.30)	0.353	0.96 (0.79, 1.16)	0.687
Sex				
Female	1		1	
Male	1.34 (1.20, 1.49)	<0.001	1.53 (1.36, 1.71)	<0.001
Age category (years) ^a^	1.04 (1.03, 1.04)	<0.001	1.04 (1.04, 1.05)	<0.001
Primary race:				
White or Caucasian	1		1	
American Indian or Alaskan Native	1.09 (0.06, 7.42)	0.936	3.04 (0.15, 21.5)	0.328
Asian	1.27 (1.00, 1.59)	0.047	1.54 (1.20, 1.97)	<0.001
Black or African American	0.86 (0.67, 1.10)	0.236	1.20 (0.91, 1.56)	0.191
Hispanic or Latino	0.86 (0.76, 0.98)	0.018	1.49 (1.28, 1.73)	<0.001
Native Hawaiian or Other Pacific Islander	0.56 (0.23, 1.15)	0.148	0.98 (0.40, 2.08)	0.968
Other	1.12 (0.91, 1.38)	0.261	1.30 (1.05, 1.61)	0.015
Unknown	1.52 (1.04, 2.18)	0.026	2.04 (1.37, 2.99)	<0.001
Facility Location Name:				
Providence Holy Cross Medical Center	1		1	
Providence Little Company of Mary (San Pedro)	0.80 (6.1, 1.04)	0.107	0.87 (0.65, 1.14)	0.318
Providence Little Company of Mary (Torrance)	0.95 (0.82, 1.11)	0.516	0.95 (0.81, 1.12)	0.570
Providence Saint John’s Health Center	1.68 (1.38, 2.04)	<0.001	1.58 (1.27, 1.95)	<0.001
Providence St. Joseph Medical Center (Burbank)	0.97 (8.3, 1.14)	0.720	0.93 (0.78, 1.11)	0.411
Providence Tarzana Medical Center	1.06 (0.88, 1.27)	0.549	1.01 (0.83, 1.23)	0.934

CI = confidence interval, OR = odds ratio, SSRI/SNRI = selective serotonin reuptake inhibitor/selective norepinephrine reuptake inhibitor. ^a^ = reference age category is <18 years.

## Data Availability

The full dataset is available from the corresponding author, upon motivated request.

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
