# Peer review of "Ongoing Use of SSRIs Does Not Alter Outcome in Hospitalized COVID-19 Patients: A Retrospective Analysis"

_jcm, 2021, doi:10.3390/jcm11010070_

Round 1
Reviewer 1 Report
This is a review of the manuscript “Ongoing Use of SSRIs and the Hospital Course of COVID-19 Patients: A Retrospective Outcome Analysis” submitted for publication in The Journal of Clinical Medicine. This an interesting observational retrospective study, aiming at replicating prior work that observed a potential association between antidepressant use and reduced mortality in patients hospitalized for COVID-19. On a more general, and potentially more innovative front, it is one of a small (but quickly increasing) number of studies looking at this topic, which is an important area of inquiry with potential major implications. Although this study has the potential to add to the literature, I think that this submission suffers from several methodological flaws that should be addressed before it can be suitable for publication.
1/ Title: The title should reflect that the outcome studied in this report is mortality.
Introduction
2/ The background section needs to be updated ; as it stands, it could give the readers the impression that potential effects of SSRIs against COVID-19 mainly rely on press releases. I listed below the studies that I think should be cited and briefly discussed in the introduction (some of them being already cited in the discussion by the authors), before the authors can formulate an hypothesis:
- a) Anti-inflammatory effects of several SSRIs are already known and supported by (i) a meta-analysis of studies conducted among depressed people (Köhler et al. Molecular neurobiology, 2018, 55(5):4195-4206) showing that antidepressant use reduces levels of proinflammatory cytokines such as IL-6, TNFalpha, IL-10 and CCL-2 (which are inflammatory markers involved in severe COVID-19) and (ii) two animal studies (Roumestan C et al. Respir Res, 2007;8(1):35 and Rosen DA et al. Sci Transl Med. 2019;11(478):eaau5266) showing that fluoxetine and fluvoxamine are associated with reduced mortality and decreased inflammation in LPS-induced endotoxic septic shock mouse models.
- b) Four published clinical trials (https://pubmed.ncbi.nlm.nih.gov/34719789/; https://jamanetwork.com/journals/jama/fullarticle/2773108; https://www.ncbi.nlm.nih.gov/pmc/articles/PMC7888564/; https://www.ncbi.nlm.nih.gov/pmc/articles/PMC8550952/), including two randomized placebo-controlled trials, have shown beneficial effects of fluvoxamine in COVID-19 patients (3 among outpatients and 1 among ICU patients).
- c) The observational study of Hoertel et al. should be better detailed, particularly the sample size and the adjustments variables used to allow readers compare them with those used (and not used) in the present study. This information may be sufficient to understand the discrepancy in results between this study and the prior one.
- d) Results from that study have been confirmed by other observational studies also involving hospitalized patients with COVID-19 and taking into account risk factors for COVID-19 mortality: (i) Diez-Quevedo et al.. Acta Psychiatr Scand. 2021 Jun;143(6):526-534 in which there was a significant association between antidepressant use (N=481) and reduced mortality among 2,150 in patients hospitalized with COVID-19 and (ii) a preprint of Németh Z et al. available at SSRN 3896539 showing a significant association between fluoxetine use (N=110) and reduced 28-day mortality among 269 in patients hospitalized with COVID-19. These papers should be cited.
- e) Finally, numerous preclinical studies (Carpinteiro et al., Cell Report Med, 2020; Zimniak, […], Bodem. Sci Rep. 2021; Dechaumes et al. Microorganisms. 2021; Schloer et al. Br J Pharmacol. 2021; Schloer S, et al. Emerg Microbes Infect. 2020 ; Senem Merve et al., https://doi.org/10.1101/2021.03.22.436379) support an in vitro efficacy of several SSRIs, particularly fluoxetine, against SARS-CoV-2, with different types of cells and different variants.
3/ Based on these prior studies, authors should formulate an hypothesis at the end of the introduction.
Method
4/ Please clarify whether inclusion and exclusion criteria rely on “any antidepressant” or “any SSRI”.
If the later, it should be checked whether patients from the control group could be exposed to non-SSRI antidepressant. It could be a problem as Hoertel et al. study suggests that both SSRIs and non-SSRI antidepressants could be associated with decreased mortality in this population. Ideally, including all patients with any antidepressant (both SSRI and non-SSRI antidepressants) would be the best approach given the conclusion of the Hoertel et al. study and could increase statistical power of that study.
5/ Given the multicenter design (6 hospitals), potential center effect should be checked, and the variable “center” should be taken into account in the analysis.
6/ The inclusion period is wide (from 03/2020 to 03/2021), and care is likely to have progressed with time. Particularly, the use of certain treatments in the more recent months (dexamethasone, monoclonal antibodies, etc.) is likely to have improved outcomes of hospitalized patients with COVID-19. If patients from the exposed group were more likely to have been hospitalized early and not to have benefited from those treatments, it may result to bias results towards the null hypothesis. It would thus be important to also take into account time of hospitalization/period of “inclusion” in the analysis. In addition, it would be important, if possible, to also take into account whether patients were part of a clinical trial testing a COVID-19 medication or were prescribed dexamethasone or monoclonal antibodies at any time during their hospitalization.
7/ Could the mean dose (fluoxetine-equivalent dose) of SSRIs/antidepressants calculated in this sample? Lower dose may also be a source of discrepancy between results from that study and prior ones.
8/ Importantly, study results potentially suffer from two major biases related to observational studies that are not addressed in the current analysis: indication bias (psychiatric disorders) and bias by confounding (it is well-known that medical comorbidities and obesity are more prevalent in people taking antidepressants (https://www.cambridge.org/core/journals/the-british-journal-of-psychiatry/article/medical-disorders-in-people-with-recurrent-depression/405B95A0386565858B160CAE3E109AF5), and particularly in those hospitalized for COVID if these treatments are effectively beneficial, and extensive literature has already showed that medical comorbidities and obesity, along with age, are central risk factors for COVID-related mortality). The analyses should at least try to address these two potential major biases by adjusting for medical comorbidities and psychiatric disorders. Finding non-significant differences without adjusting for these factors is already surprising and should be discussed.
9/ Therefore, giving the design of the study and the observed very high rates of medical comorbidities in inpatients with COVID-19 taking antidepressants in Hoertel et al. study, I think it is very important to build a matched analytic sample and/or to adjust in the full sample, based on (i) age, sex, race/ethnicity, as well as (ii) the total number of comorbid medical disorders, (iii) obesity, (iv) period of inclusion, (v) center, (vi) if possible, specific treatments for COVID such as monoclonal antibodies or dexamethasone or inclusion as part as COVID-19 clinical trial, and factors related to indication bias, including (vii) a diagnosis of mood or anxiety disorder, (viii) of dementia or psychotic disorder, (ix) or of any other psychiatric disorder, and the frequent co-prescription of medications in individuals taking antidepressants that are likely to increase mortality risk, including (x) any benzodiazepine or Z-drug and (xi) any antipsychotic medication.
Indication bias and co-prescribed medications are particularly important methodological issues to address in this study.
10/ I think that if most of these variables are included in this analysis, then the authors could say that they “recreated the original French study”. If not, I think that the main conclusion of this article should better be that they simply do not have the data to examine this issue.
11/ Power calculation is necessary to be able to interpret the results. Specifically, what is the power of this study to detect the difference observed by the study of Hoertel et al. if it exists, and what is the statistical power to detect a moderate effect size? It should be reported in the results and discussion sections.
12/ A detailed table of the frequency of each SSRI molecules in the exposed group should be provided.
13/ Exploratory analyses following the same set of adjustments should be performed at the molecule level (e.g., fluoxetine, paroxetine, etc…) to examine the associations of each molecule with mortality, as prior studies indicate variability in potential efficacy across SSRIs.
14/ The authors should also report multicollinearity diagnostics, assess the fit of the data, and examine the potential influence of outliers.
15/ The associations between each characteristic listed in comment #9 and mortality should be presented in supplementary material.
16/ “A limitation of our study is that our patient population did not have any patients 158 taking fluvoxamine and this is in direct contrast to the French study and to an open label, 159 prospective cohort study of fluvoxamine”. In fact, only 1 patient was under fluvoxamine in the French study (out of 345 patients under antidepressants), so fluvoxamine is unlikely to have explained these results. Please correct this sentence.
17/ The paragraph on S1R should specify that fluvoxamine may have the strongest binding affinity to S1R of all the SSRIs in rat brain membranes. However, it should be noted that many antidepressants do have anti-inflammatory properties at usual doses observed in depressed patients as shown in Köhler et al. Molecular neurobiology, 2018, 55(5):4195-4206. and that fluoxetine (Roumestan C et al. Respir Res, 2007;8(1):35) is associated with reduced mortality and decreased inflammation in a LPS-induced endotoxic septic shock mouse model.
18/ The last part of the discussion on other potential medications (statins etc…), not examined in this report, is speculative and should be removed.
19/ Among the potential antiviral and anti-inflammatory mechanisms that may underlie the potential effect of fluvoxamine, inhibition of the ASM/ceramide system (https://www.ncbi.nlm.nih.gov/pmc/articles/PMC7598530/ ; https://www.ncbi.nlm.nih.gov/pmc/articles/PMC8488928/ ; https://www.ncbi.nlm.nih.gov/pmc/articles/PMC8359627/; https://ascpt.onlinelibrary.wiley.com/doi/10.1002/cpt.2317 ; https://www.mdpi.com/1422-0067/22/19/10198 ; https://www.mdpi.com/1422-0067/22/9/4794) seems to have substantial biological evidence. These studies should also be briefly discussed.
20/ “In this retrospective study of 9,044 patients hospitalized for COVID-19 in 6 California hospitals of a large hospital system in the Western US, prior use of SSRIs or SNRIs did not reduce mortality”. I thought that only SSRIs were included in this analysis. Please clarify.
21/ I think that the limitations section should include the points detailed above.
22/ In case of non-significant results following appropriate adjustments, how all prior results and findings from this study can be reconciliated and explained?
Conclusion
23/ I think that the discussion/conclusion should be substantially toned down (e.g. “as a result of this study, the use of the SSRI /SNRI drug class does not hold a particular advantage”) given the potential important methodological issues of the current submission.
Author Response
We appeal the comments of Reviewer 1. Reviewer 1 has greatly overstepped the bounds of an appropriate review and has not been respectful of our work. 23 comments are provided and it is not be possible to respond to them as this would require us to completely redo the paper and we would be writing this reviewer’s paper and not our own. The reviewer is unreasonable and wants us to follow instructions as to how we should collect our data and which specific publications to cite. We could not possibly satisfy this reviewer.

Reviewer 2 Report
First of all, I would like to congratulate the article team for the work done. I believe that all the information is very well exposed, the literature well reviewed and follows the scientific method rigorously.
I think this work could be published with a few minor changes.
INTRODUCTION
I would include at least one publication on psychotropic drugs in general and their effects on evolution and mortality in patients with COVID-19. I recommend this Spanish article carried out in a population of 2150 hospitalized patients: Diez-Quevedo et al. Mental disorders, psychopharmacological treatments, and mortality in 2150 COVID-19 Spanish inpatients. Acta Psychiatr Scand. 2021 Jun;143(6):526-534. doi: 10.1111/acps.13304.
At the beginning of line 39: Various psychopharmacological treatments, such as anxiolytics, hypnotics and antidepressants have been independently associated with lower mortality risk in patients hospitalized for COVID-19.
MATERIALS, METHODS AND RESULTS
In the Materials and Methods section, it is explained that several variables have been obtained from the medical records, including length of stay, oxygen saturation upon admission,… among others (lines 83-87). However, these variables have not been taken into account in the development of the analyzes. Given the significant comorbidity of patients admitted for COVID-19, I think it would be very important to include concomitant somatic pathologies as adjustment variables in the analyzes to see if they directly influence the results obtained. Especially those diseases that have been directly associated with high mortality from COVID-19 (dyslipidemia, obesity, arterial hypertension, diabetes mellitus, ischemic heart disease,…).
It would also be important to see if SSRIs have had a statistically significant association with the risk of mortality or complications in less serious patients (not admitted to the ICU).
If these data are available, as shown in the methods and materials section, it would be very interesting to be able to classify subgroups of patients in whom a benefit could be obtained with the use of SSRIs.
Author Response
We thank the reviewer for thoroughly scrutinizing our manuscript. As requested, we have revised the manuscript and addressed the specific comments of Reviewer #2. The revised sections are delineated in red in a marked copy of the manuscript text.
Below, we provide a point-by-point response to this reviewer’s comments.
- COMMENT: First of all, I would like to congratulate the article team for the work done. I believe that all the information is very well exposed, the literature well reviewed and follows the scientific method rigorously. I think this work could be published with a few minor changes.
RESPONSE: We thank the reviewer for this positive assessment.
- COMMENT: INTRODUCTION
I would include at least one publication on psychotropic drugs in general and their effects on evolution and mortality in patients with COVID-19. I recommend this Spanish article carried out in a population of 2150 hospitalized patients: Diez-Quevedo et al. Mental disorders, psychopharmacological treatments, and mortality in 2150 COVID-19 Spanish inpatients. Acta Psychiatr Scand. 2021 Jun;143(6):526-534. doi: 10.1111/acps.13304.
At the beginning of line 39: Various psychopharmacological treatments, such as anxiolytics, hypnotics and antidepressants have been independently associated with lower mortality risk in patients hospitalized for COVID-19.
RESPONSE: We agree with the need to cite publications on the general category of psychotropic drugs in COVID-19 and their effects on disease course and mortality. We have cited the Diez-Quevedo reference and several others and expanded the discussion.
- COMMENT: MATERIALS, METHODS AND RESULTS
In the Materials and Methods section, it is explained that several variables have been obtained from the medical records, including length of stay, oxygen saturation upon admission,… among others (lines 83-87). However, these variables have not been taken into account in the development of the analyzes. Given the significant comorbidity of patients admitted for COVID-19, I think it would be very important to include concomitant somatic pathologies as adjustment variables in the analyzes to see if they directly influence the results obtained. Especially those diseases that have been directly associated with high mortality from COVID-19 (dyslipidemia, obesity, arterial hypertension, diabetes mellitus, ischemic heart disease,…).
RESPONSE: We have included only those variables that were collected and included in the analyses. Unfortunately, at this time, we do not have comorbidity data available for those not taking SSRIs. We will conduct this in the future once data is obtained.
- COMMENT: It would also be important to see if SSRIs have had a statistically significant association with the risk of mortality or complications in less serious patients (not admitted to the ICU). If these data are available, as shown in the methods and materials section, it would be very interesting to be able to classify subgroups of patients in whom a benefit could be obtained with the use of SSRIs.
RESPONSE: This particular study focuses on seriously ill patients. Data on non-ICU patients is not available to us. Almost all COVID-19 deaths take place in the ICU. The less severe cases remain on the inpatient floor (general ward). It is certainly possible one of those patients could die but the overwhelming number of COVID-19 patients in an acute care facility are transferred to the ICU before death. A future study may look at patients not admitted to ICU. We also reference a new study by Clelland et al (Reference # 41) that looks at whether people on SSRIs are more likely to test positive for COVID-19. However, it does not address the course or severity.
We thank the reviewer and believe that the manuscript is improved as a result of their input. We hope you will agree, and decide in favor of accepting our report at this time.

Reviewer 3 Report
The authors carried out the study on impressive material. But I have recommendations to improve the article.
Introduction
There is absolutely no data on the pathogenesis of inflammation and how SSRIs relate to it. I recommend that the authors include a few sentences of clarification. I think we need to expand on how serotonin metabolism can influence peripheral inflammation with COVID-19
Materials and methods
There is no evidence that information on antidepressants was collected. Judging from the "Discussion" section, the authors know that no one was taking fluvoxamine, So, information on the SSRIs received should be added.
Results
In Table 1 I recommend showing the significance of differences in demographic parameters between the groups.
Add results of analysis of patient mortality by SSRI taken. In the French studies, fluvoxamine was studied. The significance of this study can be increased by presenting the analysis for each antidepressant.
Discussion
The discussion about fluvoxamine should partly be moved to the introduction. It would then be appropriate to do a separate analysis of SSRIs in the sample of this study.
If I were the authors I would avoid the judgement that the absence of fluvoxamine affects their results. As their reasoning suggests, its effect on the outcome of COVID-19 has not been proven in the high-quality studies.
The reasoning about the benefit of keeping SSRIs in the intensive care unit has no evidence. These conclusions do not follow from the results of this study. I recommend that the authors change their style and state this in a presumptive way.The authors do not cite studies that compare withdrawal or continuation of SSRIs in the intensive care unit.
Conclusion
The conclusions are grossly inconsistent with the results obtained. The authors' recommendations for the use of fluvoxamine should be excluded, as the drug was not available in the present study. All assumptions about fluvoxamine, and the synergistic effects of other drugs, should either be deleted or left in the discussion section. But I think it is inappropriate to focus a lot of attention on things that have not been explored in this study.
I recommend focusing on the findings as they have value in their own right.
I also recommend that the authors put the main conclusion in the title of the article.
Author Response
We thank the reviewer for thoroughly scrutinizing our manuscript. As requested, we have revised the manuscript and addressed the specific comments of Reviewer #3. The revised sections are delineated in red in a marked copy of the manuscript text.
Below, we provide a point-by-point response to this reviewer’s comments.
- COMMENT: Introduction
There is absolutely no data on the pathogenesis of inflammation and how SSRIs relate to it. I recommend that the authors include a few sentences of clarification. I think we need to expand on how serotonin metabolism can influence peripheral inflammation with COVID-19
RESPONSE: We have added this discussion with references on page 2.
- COMMENT: Materials and methods
There is no evidence that information on antidepressants was collected. Judging from the "Discussion" section, the authors know that no one was taking fluvoxamine, So, information on the SSRIs received should be added.
RESPONSE: We have added information on types of anti-depressants taken.
- COMMENT: Results
In Table 1 I recommend showing the significance of differences in demographic parameters between the groups.
RESPONSE: Since this is just a descriptive table, we did not see a need to add p-values.
- COMMENT: Add results of analysis of patient mortality by SSRI taken. In the French studies, fluvoxamine was studied. The significance of this study can be increased by presenting the analysis for each antidepressant.
RESPONSE: There is a very uneven distribution of the individual SSRIs taken (that’s why the data was collapsed into a yes/no variable) so it is not feasible to do this analysis. If we can increase our numbers of the infrequent anti-depressants in the future, then we can do this analysis.
- COMMENT: Discussion
The discussion about fluvoxamine should partly be moved to the introduction. It would then be appropriate to do a separate analysis of SSRIs in the sample of this study.
RESPONSE: We have removed much of the discussion of fluvoxamine, but have still referred to this drug because that is the only SSRI investigated to any depth in the literature and, although we made it clear that we had no patients on fluvoxamine in our study, the literature in this area is still quite relevant.
- COMMENT: If I were the authors I would avoid the judgement that the absence of fluvoxamine affects their results. As their reasoning suggests, its effect on the outcome of COVID-19 has not been proven in the high-quality studies.
RESPONSE: We now avoid this judgement
- COMMENT: The reasoning about the benefit of keeping SSRIs in the intensive care unit has no evidence. These conclusions do not follow from the results of this study. I recommend that the authors change their style and state this in a presumptive way. The authors do not cite studies that compare withdrawal or continuation of SSRIs in the intensive care unit.
RESPONSE: On page 9, we now state clearly that “Our study does not provide evidence that there is any benefit to keeping patients on SSRIs in the ICU.”
- COMMENT: Conclusion
The conclusions are grossly inconsistent with the results obtained. The authors' recommendations for the use of fluvoxamine should be excluded, as the drug was not available in the present study. All assumptions about fluvoxamine, and the synergistic effects of other drugs, should either be deleted or left in the discussion section. But I think it is inappropriate to focus a lot of attention on things that have not been explored in this study.
RESPONSE: We have rewritten substantial parts of this section.
- COMMENT: I recommend focusing on the findings as they have value in their own right.
RESPONSE: We appreciate this approach and have put greater emphasis on the findings of our study.
- COMMENT: I also recommend that the authors put the main conclusion in the title of the article.
RESPONSE: We have changed the title to reflect the conclusions. New title: “Ongoing Use of SSRIs Does Not Alter Outcome in Hospitalized COVID-19 Patients: A Retrospective Analysis”
We thank the reviewer and believe that the manuscript is improved as a result of their input. We hope you will agree, and decide in favor of accepting our report at this time.

Round 2
Reviewer 1 Report
This is a review of the revision of the manuscript “Ongoing Use of SSRIs and the Hospital Course of COVID-19 Patients: A Retrospective Outcome Analysis”.
The role of a reviewer is to ensure that the methods and analyses are adequately detailed and performed and the limitations adequately indicated to allow the readers to judge the scientific merit of the study, and that the article cites relevant work done by other scientists. A reviewer shows respect to an author by spending time in reading the full manuscript and supplementary material and by writing an honest report with precise suggestions, that authors are free to accept or refute based on scientific evidence. An author shows respect to the reviewer by providing a response (even short) to her/his comments, even when he/she chooses to reject a suggestion.
While several of my comments appear to have been adequately addressed in the revision, I still remain concerned about two biases in the current analyses that have not been addressed: indication bias, confounding due to the non-inclusion of medical comorbidities (including obesity) and co-prescribed benzodiazepines and antipsychotics.
- 1/ Indication bias: patients do not take antidepressants at random in this observational study, but for specific medical indications including possibly depressive or anxiety disorders, eating disorders, behavioral problems occurring during dementia or neuropathic pain, conditions that may for some of them increase the risk of COVID-19-related mortality according to prior studies (Acta Psychiatr Scand 2021, 143, 526-534; https://www.nature.com/articles/s41586-020-2521-4; Vai B et al., Lancet Psychiatry. 2021 Sep;8(9):797-812, even if this last study mostly included associations adjusted only for age and sex).
- 2/ Confounding: it is well-known that psychiatric disorders (and thus use of psychotropic medications due to psychiatric disorders) are associated with greater rates of a broad range of medical conditions (https://www.bmj.com/content/365/bmj.l1577.short), including cardiovascular diseases and obesity, which are associated with greater COVID-19-related mortality (https://www.nature.com/articles/s41586-020-2521-4), and especially in inpatients hospitalized for COVID-19 (doi: 10.1038/s41380-021-01393-7). Therefore the statement L217 “Since use of SSRIs for anxiety, OCD or depression seems unrelated to other comorbidities known to affect COVID morbidity and mortality (hypertension, diabetes, heart disease), direct impact of SSRIs on the risk factors for severe COVID-19 is minimal” is speculative and untrue.
- 3/ Benzodiazepines and antipsychotics, medications that are frequently co-prescribed with antidepressants, have been reported as potentially associated with greater COVID-19-related mortality (Vai B et al., Lancet Psychiatry. 2021 Sep;8(9):797-812).
Because the AOR between SSRIs/SNRIs (or SSRIs only?) and mortality was non-significantly lower than 1 after adjusting for age, gender, race and facility, and since it is possible/likely that patients under SSRIs/SNRIs have greater rates of medical comorbidities and obesity, diseases for which the antidepressant was prescribed, and co-prescriptions of benzodiazepines and antipsychotics, which all have been associated with higher risk of death based on studies cited above, the authors cannot rule out with their current analyses a potential protective association.
In case of non-inclusion of these key parameters in their analyses, these limitations should at least be clearly acknowledged, discussed as a possible source of discrepancies between results from this study and those of several prior studies that were able to adjust, at least partly, for these parameters.
Finally, sentences such as “As a result of this study, the use of the SSRI /SNRI drug class does not hold a particular advantage in the patient population already taking these drugs” or “Our study does not provide evidence that there is any benefit to keeping patients on SSRIs in the ICU” might favor the stop of these treatments by clinicians, despite 4 clinical trials, including 2 randomized trials, and several observational studies supporting a potential benefit of several antidepressants in these patients, and despite the potential risk of aggravation of the psychiatric conditions that have led to this prescription.
I strongly encourage authors to address these comments and also suggest more careful conclusions given the observational design of the study and the potential biases detailed above in the current analyses.
Author Response
We have addressed his concerns in revision. We do not have detailed data on additional comorbidity . Also, we do not have detailed data on specific psychiatric diagnosis of each patient in study.
Also, we have addressed issue of continuation and discontinuation of SSRIs in the ICU. This was also noted in other reviewer’s comments.
Thus, we have addressed reviewer 1 despite the fact that we found feedback inappropriate and burdensome.
Reviewer 3 Report
I have no comments
Author Response
Thank you.